# Associations between physical activity and asthma, eczema and obesity in children aged 12–16: an observational cohort study

Russell Jago,[1,2] Ruth E Salway,[1] Andy R Ness,[3,4] Julian P Hamilton Shield,[3] Matthew J Ridd,[5] A John Henderson[6]

For numbered affiliations see end of article.

**Correspondence to**
Dr Russell Jago;
russ.jago@bristol.ac.uk

## ABSTRACT

**Objectives** To compare the physical activity of adolescents with three common long-term conditions (asthma, eczema and obesity) with adolescents without these conditions.

**Design** Cross-sectional and longitudinal analyses of adolescents at ages 12, 14 and 16 in a large UK cohort study.

**Setting** The Avon Longitudinal Study of Parents and Children.

**Participants** 6473 adolescents with complete accelerometer data at at least one time point.

**Methods** Mean minutes of moderate to vigorous intensity physical activity (MVPA) and sedentary time per day were derived from accelerometer-based measurements at ages 12, 14 and 16. Obesity was defined at each time point from height and weight measurements. Parents reported doctor-assessed asthma or eczema. Cross-sectional and longitudinal regression models examined any differences in MVPA or sedentary time for adolescents with asthma, eczema or obesity compared with those without.

**Results** In longitudinal models, boys engaged in an average of 69.7 (95% CI 67.6 to 71.7) min MVPA at age 12, declining by 3.1 (95% CI 2.6 to 3.6) min/year while girls' average MVPA was 47.5 (95% CI 46.1 to 48.9) min at age 12, declining by 1.8 (95% CI 1.5 to 2.1) min/year. There was no strong evidence of differences in physical activity patterns of those with and without asthma or eczema. Obese boys engaged in 11.1 (95% CI 8.7 to 13.6) fewer minutes of MVPA, and obese girls in 5.0 (95% CI 3.3 to 6.8) fewer minutes than their non-obese counterparts. Cross-sectional models showed comparable findings.

**Conclusions** Mean minutes of MVPA per day did not differ between adolescents with asthma or eczema and those without, but obese adolescents engaged in fewer minutes of MVPA. Findings reinforce the need for strategies to help obese adolescents be more active but suggest no need to develop bespoke physical activity strategies for adolescents with mild asthma or eczema.

## INTRODUCTION

Obesity, eczema (synonyms: atopic eczema/dermatitis) and asthma are three of the most prevalent childhood long-term conditions in

the UK. Data from the 2015/2016 National Child Measurement Programme showed that 9% of 4–5-year-olds and 20% of 10–11-year-olds in England were obese.[1] It has been estimated that 20% of children are diagnosed with eczema[2] and 9% with asthma,[3] and children with eczema often have asthma. These long-term conditions place substantial burden on quality of life and healthcare expenditure. Understanding factors associated with the prevalence of these conditions is important to aid the development of prevention and management strategies.

Physical activity is a modifiable lifestyle factor that has been associated with adiposity[4] and asthma[5] in children, and adults with asthma engage in lower levels of physical activity than those without.[6] Low levels of physical activity reduce the quality of life in childhood and increase the risk of other comorbidities.[7] Physical inactivity may be a problem among children with long-term conditions. There is some evidence that physical activity levels are lower among children with asthma than children without asthma.[8]

Children with eczema may be less active than children without eczema due to reluctance to sweat which can irritate eczematous skin, but the current evidence base is very weak.[9] It may be that tailored strategies are needed to help those with asthma and eczema be more physically active, but the development of such approaches would only be warranted if there were evidence of lower physical activity.

There is some evidence to show that physical activity levels are lower among children who are overweight or obese.[10] While there have been many studies that have looked at the association between physical activity and obesity, these have tended to focus on cross-sectional surveys, or national surveys that have looked at associations over time but not within participants over time, for which cohort studies are essential. An analysis of the Avon Longitudinal Study of Parents and Children (ALSPAC) cohort reported a graded inverse association between physical activity and obesity at age 12[11]; an increase of 15 min of moderate to vigorous intensity physical activity (MVPA) per day was associated with 10% lower fat mass in girls and 12% lower in boys.[12] However, the authors used an ALSPAC-specific definition of MVPA[13] and reported an average MVPA of 20 min per day while the majority of current studies use the Evenson cut-points[14] which a comparison study reported was the reliable threshold[15] for young people. This difference is important as other studies that use Evenson cut-points typically report MVPA in the range of 50 to 70 min per day[16–18] and, as such, it is important to have data from ALSPAC that have been processed using the Evenson cut-points to facilitate international comparisons.

The overall objective of this paper is to examine the physical activity of adolescents with three of the most common long-term conditions and if these associations change as young people move through adolescence, using objective assessments of physical activity. This evidence is needed to identify whether there are important differences between children with different long-term conditions. The specific aims were: (1) to compare mean time spent sedentary and engaged in MVPA for adolescents with asthma, eczema and obesity at ages 12, 14 and 16; and (2) to examine prospective associations between sedentary and MVPA time from age 12 to 16 for those with and without these conditions.

## METHODS

ALSPAC is a birth cohort study based in the former county of Avon (UK).[19–21] A total of 15 247 pregnant women with expected delivery dates between April 1991 and December 1992 were recruited, with 14 701 children alive at 1 year of age. Ethical approval for the study was obtained, and written informed consent was obtained for all mothers, with parents providing written informed consent for young people's participation. Please note that the study website contains details of all the data that are available through a fully searchable data dictionary (http://www.bris.ac.uk/alspac/researchers/data-access/data-dictionary/).

### Accelerometer data

Adolescents wore an Actigraph AM7164 (Actigraph LLC, Fort Walton Beach, Florida, USA) accelerometer on their hip for 7 consecutive days. The first accelerometer measurement was taken at an average age of 11 years and 9 months in 2003–2004, with subsequent measurements at ages 13 years and 10 months, and at 15 years and 6 months. Accelerometer data were processed using Kinesoft (V.3.3.75; Kinesoft, Saskatchewan, Canada) and analysis was restricted to those who provided at least 3 days of valid data. A valid day was defined as at least 500 min of data, after excluding intervals of ≥60 min of zero counts allowing up to 2 min of interruptions. The average number of MVPA (≥2296 counts per minute (CPM)) and sedentary minutes per day (<100 CPM) were derived for each adolescent based on the Evenson thresholds.[14] Accelerometer data with over 11715 CPM, fewer than 10 sedentary minutes or zero MVPA were considered erroneous and excluded from the analysis (16, 41 and 21 respondents at ages 12, 14 and 16, respectively).[22]

### Exposures

Height and weight were measured at the same time as the accelerometer data in a clinic setting by trained fieldworkers. Body mass index (BMI) was used to create age-specific and sex-specific overweight and obesity indicators using 85th and 95th percentiles based on UK BMI reference charts.[23] Throughout this paper, the overweight category comprises those participants who were overweight but not the participants who were obese, who are defined as obese throughout. The mother was asked 'Has a doctor ever told you that your child has asthma?' at ages 8, 11 and 14, and 'Has a doctor ever told you that your child has eczema?' at ages 11 and 14. These were combined to form lifetime doctor-diagnosed asthma and eczema prevalence: 'has asthma ever been reported as diagnosed by a doctor?' and 'has eczema ever been reported as diagnosed by a doctor?', assessed by ages 11 and 14. As these ages do not correspond to accelerometer collection, current asthma/eczema assessments were unavailable.

### Confounders

The child's sex as recorded at birth and the age of the child at the accelerometer data collection were included as confounders. The mother's highest education variable was created by combining from data at 32 w gestation and age 8 to provide the highest education recorded in the data set. Household social class was calculated as the highest occupational social class reported by mother or partner from data at 18 weeks' gestation (both) and ages 4 (mother) and 8 (partner). Puberty was assessed from yearly questionnaires. For girls, an indicator of onset of menarche was used. For boys, Tanner stages based on pubic hair was used, as self-report data on Tanner stages for genitals has been found to be unreliable.[24]

## Patient and public involvement

The concept for this paper was informed by previous research with children with asthma and their parents who expressed challenges in relation to being physically active.[25] As this was a secondary analysis of data that were collected 10 years ago, participants were not involved in the design of the specific study or study recruitment which were conducted as part of the ALSPAC cohort. Results will be shared with ALSPAC participants via the study website.

## Statistical analysis

Cross-sectional data were analysed at three timepoints corresponding to accelerometer data at average ages of 12, 14 and 16. We included any child with at least 3 valid days of accelerometer data. Obesity and overweight were calculated at the same timepoints. Asthma and eczema diagnoses did not correspond to the same timepoints, and so were taken from the closest prior measurement, to record lifetime diagnosis up to the point of the accelerometer measurement. Mother's highest education and household social class were calculated as described above and the same value used for all analyses. Cross-sectional analysis was based on 5735, 4078 and 2198 cases for ages 12, 14 and 16, respectively. For the longitudinal analyses, we combined data from all three timepoints, resulting in 6473 cases with data for at least one timepoint, and 1619 (25%) at all three.

Multiple imputation was used to account for missing data. For the cross-sectional analyses, data were imputed for each data set separately. For the longitudinal analysis, data were imputed if the child had accelerometer data for that timepoint, but not between timepoints, although data from other timepoints was used to impute missing data. Multiple imputation methods were used to create 20 imputed data sets at each timepoint for boys and girls separately, using 20 cycles of regression switching and combined regression coefficients across data sets using Rubin's rules.[26] All exposures, outcomes and confounders were included.

### Cross-sectional analyses

For each age, characteristics were described separately for all adolescents, those with asthma, eczema, who were overweight or obese, and for adolescents with none of the conditions. Average minutes of MVPA and sedentary time were compared between those with long-term conditions and those without using t-tests. Regression models were fitted for MVPA and sedentary minutes for all adolescents, and for boys and girls separately. Models include the asthma, eczema, overweight and obese indicators, and child age, sex, mother's education and household social class as confounders. For comparative purposes, minimally adjusted models which adjusted for just child age and sex were also run and are shown in online supplementary table.

### Longitudinal analysis

Random effect models were fitted to identify changes in physical activity measures over time between the long-term conditions, accounting for individual variability in measurements. As the cross-sectional analyses suggest different patterns among boys and girls, all models were run separately by sex. We included indicators for asthma, eczema, obesity and overweight, and age at time of clinic to capture linear changes over time. All models were adjusted for mother's highest education and household social class. In this model, the constant term captures the average physical activity of a healthy child (ie, with none of the long-term conditions) at age 12 for average confounder values. The age coefficient estimates a linear change in physical activity (or sedentary time) over time, which is the same for all long-term conditions, and additional terms for each health condition reflect any difference in the baseline activity.

A model was also fitted to assess if there were interactions between the long-term conditions to assess whether there was any additional effect associated with having multiple conditions. Finally, a model which allowed the change in physical activity over time to depend on health condition; so, for example, MVPA for an adolescent with asthma might decrease at a different rate between ages 12 and 16 than for an adolescent without these coniditons was run. As self-reported pubertal status is susceptible to error[27] and pubertal status could be linked to obesity,[28] pubertal status was considered in a secondary sensitivity analysis. All analyses were performed in Stata V.15.0.[29]

## RESULTS

Online supplementary table S1 summarises the characteristics of the imputed data, with summaries for observed data reported in online supplementary table S2. Missing data ranged from <1% (obesity) to 15% (eczema). Between 24% and 27% of adolescents had ever been diagnosed with asthma, between 21% and 27% had ever been diagnosed with eczema and between 13% and 15% were obese. Between 13% and 15% had more than one condition, with the most common combination being asthma and eczema (between 7% and 9% of adolescents). The combinations at each age are summarised graphically in online supplementary figure S1. The average minutes of MVPA per day decreased from 57 min at age 12 to 47 min at age 16, with average sedentary minutes increasing from 354 min to 475 min. Online supplementary figure S2 illustrates how MVPA and sedentary time changes with age among adolescents with the different conditions.

### Cross-sectional analyses

Online supplementary table S3 summarises the difference in MVPA and sedentary time for adolescents with asthma, eczema, overweight and obese conditions, compared with those without. Tables 1 and 2 present the regression models for MVPA and sedentary time, respectively, adjusting for age, sex, mother's education and household social class, and report the difference between an adolescent with and without each health condition. These models include all conditions, including combinations;

**Table 1** Cross-sectional regression models for MVPA at average ages 12, 14 and 16

| | Age 12 (n=5735) | | | Age 14 (n=4078) | | | Age 16 (n=2198) | | |
|---|---|---|---|---|---|---|---|---|---|
| | Diff* (min) | 95 % CI | P value | Diff* (min) | 95 % CI | P value | Diff* (min) | 95 % CI | P value |
| **All†** | | | | | | | | | |
| Asthma | −1.47 | (−3.19 to 0.25) | 0.094 | −0.05 | (−2.05 to 1.95) | 0.960 | −2.62 | (−5.16 to 0.09) | 0.043 |
| Eczema | −1.85 | (−3.68 to 0.01) | 0.049 | −0.62 | (−2.69 to 1.44) | 0.554 | 0.74 | (−1.75 to 3.24) | 0.559 |
| Overweight | −7.54 | (−9.59 to 5.48) | <0.0005 | −3.12 | (−5.53 to 0.72) | 0.011 | −2.27 | (−5.55 to 1.01) | 0.175 |
| Obese | −8.94 | (−10.92 to 6.96) | <0.0005 | −7.45 | (−9.87 to 5.04) | <0.0005 | −5.29 | (−8.50 to 2.07) | 0.001 |
| **Boys†** | | | | | | | | | |
| Asthma | −2.03 | (−4.76 to 0.70) | 0.144 | 0.47 | (−2.71 to 3.65) | 0.773 | −3.22 | (−7.42 to 0.98) | 0.133 |
| Eczema | −3.52 | (−6.70 to 0.34) | 0.030 | 0.05 | (−3.51 to 3.61) | 0.977 | 2.28 | (−2.16 to 6.71) | 0.314 |
| Overweight | −11.39 | (−14.83 to 7.95) | <0.0005 | −4.47 | (−8.54 to 0.40) | 0.031 | −5.22 | (−10.95 to 0.50) | 0.074 |
| Obese | −12.52 | (−15.77 to 9.26) | <0.0005 | −10.24 | (−14.13 to 6.35) | <0.0005 | −9.42 | (−14.99 to 3.85) | 0.001 |
| **Girls†** | | | | | | | | | |
| Asthma | −1.01 | (−3.09 to 1.06) | 0.339 | −0.71 | (−3.16 to 1.75) | 0.573 | −2.48 | (−5.52 to 0.56) | 0.110 |
| Eczema | −0.49 | (−2.52 to 1.55) | 0.639 | −1.17 | (−3.75 to 1.41) | 0.372 | −0.34 | (−3.16 to 2.49) | 0.816 |
| Overweight | −4.07 | (−6.44 to 1.70) | 0.001 | −1.95 | (−4.79 to 0.88) | 0.177 | 0.19 | (−3.59 to 3.96) | 0.923 |
| Obese | −5.39 | (−7.70 to 3.07) | <0.0005 | −4.52 | (−7.49 to 1.55) | 0.003 | −1.88 | (−5.59 to 1.84) | 0.321 |

*Difference in minutes of MVPA between a child with and without the condition, with all other variables the same.
†Models include all conditions, and are adjusted for child age, sex, mother's education and household social class as confounders.
MVPA, moderate to vigorous intensity physical activity.

differences are additive if an adolescent has multiple conditions. A minimally adjusted model (see online supplementary table S4), adjusting only for child age and sex, shows very little difference to the coefficients reported in the full model.

The largest and most consistent differences are for adolescents who are overweight or obese. MVPA was 2.3–7.5 min lower among overweight adolescents, and 5.3–8.9 min lower among obese adolescents. Similarly, sedentary time was 8.6–13.8 min higher among

**Table 2** Cross-sectional regression models for sedentary time at average ages 12, 14 and 16

| | Age 12 (n=5735) | | | Age 14 (n=4078) | | | Age 16 (n=2198) | | |
|---|---|---|---|---|---|---|---|---|---|
| | Diff* (min) | 95 % CI | P value | Diff* (min) | 95 % CI | P value | Diff* (min) | 95 % CI | P value |
| **All†** | | | | | | | | | |
| Asthma | −5.14 | (−9.80 to 0.49) | 0.030 | −10.77 | (−16.90 to 4.65) | 0.001 | 0.89 | (−7.30 to 9.09) | 0.831 |
| Eczema | 0.76 | (−4.25 to 5.78) | 0.765 | 4.02 | (−2.55 to 10.59) | 0.230 | 0.95 | (−7.09 to 9.00) | 0.816 |
| Overweight | 13.82 | (8.334 to 19.30) | <0.0005 | 13.53 | (6.17 to 20.88) | <0.0005 | 8.63 | (−1.96 to 19.22) | 0.110 |
| Obese | 14.86 | (9.59 to 20.13) | <0.0005 | 22.06 | (14.68 to 29.43) | <0.0005 | 14.11 | (3.75 to 24.48) | 0.008 |
| **Boys†** | | | | | | | | | |
| Asthma | −5.86 | (−12.33 to 0.60) | 0.075 | −8.17 | (−17.15 to 0.81) | 0.074 | 3.47 | (−9.06 to 16.01) | 0.587 |
| Eczema | 2.64 | (−4.63 to 9.90) | 0.476 | 2.80 | (−7.19 to 12.80) | 0.582 | −2.30 | (−15.58 to 10.98) | 0.734 |
| Overweight | 15.07 | (7.03 to 23.12) | <0.0005 | 9.49 | (−1.78 to 20.75) | 0.099 | 16.70 | (−0.21 to 33.62) | 0.053 |
| Obese | 13.44 | (5.82 to 21.05) | 0.001 | 29.28 | (18.49 to 40.06) | <0.0005 | 21.78 | (5.32 to 38.25) | 0.010 |
| **Girls†** | | | | | | | | | |
| Asthma | −4.41 | (−11.00 to 2.19) | 0.190 | −13.26 | (−21.81 to 4.72) | 0.002 | −0.69 | (−11.54 to 10.17) | 0.901 |
| Eczema | −0.85 | (−7.67 to 5.96) | 0.806 | 4.55 | (−4.29 to 13.40) | 0.312 | 3.33 | (−6.97 to 13.63) | 0.526 |
| Overweight | 12.69 | (5.19 to 20.19) | 0.001 | 17.43 | (7.74 to 27.12) | <0.0005 | 1.87 | (−11.66 to 15.41) | 0.782 |
| Obese | 15.89 | (8.55 to 23.23) | <0.0005 | 14.44 | (4.30 to 24.58) | 0.005 | 7.63 | (−5.70 to 20.96) | 0.262 |

*Difference in minutes of sedentary time between a child with and without the condition.
†Models include all conditions, and are adjusted for child age, sex, mother's education and household social class as confounders.

**Table 3** Comparison of longitudinal models for average minutes of MVPA and sedentary time per day

| | Boys (2510 boys, 4847 measurements) | | | Girls (2744 girls, 5481 measurements) | | |
|---|---|---|---|---|---|---|
| | Estimate | 95% CI | P value | Estimate | 95% CI | P value |
| **Average MVPA minutes per day** | | | | | | |
| Baseline* (min) | 69.68 | (67.64 to 71.71) | <0.0005 | 47.52 | (46.13 to 48.90) | <0.0005 |
| Change with age (min/year) | −3.09 | (−3.578, to 2.61) | <0.0005 | −1.78 | (−2.10, to 1.45) | <0.0005 |
| Asthma† (min) | −1.24 | (−3.35 to 0.87) | 0.249 | −1.19 | (−2.81 to 0.43) | 0.150 |
| Eczema† (min) | −1.22 | (−3.62 to 1.17) | 0.316 | −0.87 | (−2.52 to 0.79) | 0.305 |
| Overweight† (min) | −8.05 | (−10.46, to 5.63) | <0.0005 | −2.33 | (−3.99, to 0.68) | 0.006 |
| Obese† (min) | −11.12 | (−13.57, to 8.68) | <0.0005 | −5.01 | (−6.78, to 3.25) | <0.0005 |
| **Average sedentary minutes per day** | | | | | | |
| Baseline* (min) | 346.64 | (341.19 to 352.08) | <0.0005 | 365.73 | (31.05 to 370.41) | <0.0005 |
| Change with age (min/year) | 28.10 | (26.94 to 29.26) | <0.0005 | 31.97 | (30.92 to 33.03) | <0.0005 |
| Asthma† (min) | −4.06 | (−9.84 to 1.73) | 0.169 | −6.45 | (−11.92 to 0.98) | 0.021 |
| Eczema† (min) | 2.16 | (−4.17 to 8.49) | 0.504 | 1.98 | (−3.54 to 7.50) | 0.482 |
| Overweight† (min) | 14.51 | (8.38 to 20.63) | <0.0005 | 12.63 | (7.14 to 18.12) | <0.0005 |
| Obese† (min) | 17.37 | (10.97 to 23.78) | <0.0005 | 15.19 | (9.28 to 21.09) | <0.0005 |

*Baseline represents the average number of minutes for a healthy individual at age 11 with mother's highest education of O level (examination at age 16) and household social class of Managerial & Technical (II).
†Change from baseline if child has this health condition.
MVPA, moderate to vigorous intensity physical activity.

overweight adolescents, and 14.1–22.1 min higher among obese adolescents. These associations tend to be slightly stronger in boys than girls. Patterns for adolescents with asthma and eczema are weaker and less consistent. Associations between MVPA and asthma were inconclusive, with a lower MVPA (2.6 min) among asthma sufferers at age 16, but no other ages, and no differences when examining boys and girl separately. There was some association between asthma and sedentary time, with lower sedentary time at ages 12 and 14 (by 5.1 min and 10.8 min, respectively), although evidence is inconclusive when considering boys and girls separately. There was very little association between eczema and either MVPA or sedentary time. We saw a slightly lower MVPA (3.5 min) among boys at age 12, but no other ages, and no associations with sedentary time. It was not possible to explore associations with combinations of long-term conditions due to small numbers (<100) for some combinations. However, there was no evidence of any additional association with having multiple conditions. So, for example, the difference between obese children with asthma and obese children without asthma is the same as the difference between non-obese children with asthma and non-obese children without.

### Longitudinal analysis

Table 3 summarises the longitudinal models for MVPA and sedentary time, for boys and girls separately. The baseline represents a typical child at age 12 with the most common mother's highest education (O level—examinations at age 16), and household class (Managerial & Technical). These coefficients are additive, so, for example,

the average number of minutes of MVPA for a 12-year-old obese boy is estimated as 69.7−11.1=58.6 min. Figure 1 plots the estimated average MVPA time and sedentary time for the different long-term conditions by age, for boys and girls.

The largest association was between physical activity and obesity, especially among boys. Asthma and eczema showed much weaker associations. A typical boy with no long-term conditions engaged in an average of 72.8 min of MVPA at age 12, declining at a rate of 3.1 min/year with age. They spent an average of 318.5 min in sedentary time at age 12, increasing at a rate of 28.1 min/year. Girls' average MVPA was lower at 49.3 min at age 12, declining at a rate of 1.8 min/year, while sedentary time was higher, with an average of 333.8 min at age 12, increasing at a rate of 32.0 min/year. We found no association between asthma and either MVPA or sedentary time for boys. Among girls, there was no association between MVPA and asthma, but asthma sufferers spent 6.5 fewer minutes in sedentary time than non-sufferers. We found no association between eczema and either MVPA or sedentary time for boys or girls.

Overweight and obese adolescents had lower MVPA and higher sedentary time. Associations are stronger for boys than for girls, and stronger for obesity than overweight. Being overweight is associated with a decrease in MVPA of 8.1 min and an increase in sedentary time of 14.5 min for boys, compared with a decrease of 2.3 min in MVPA and an increase of 12.6 min in sedentary time for girls. Obesity is associated with a decrease in MVPA of 11.1 min and an increase in sedentary time of 17.4 min

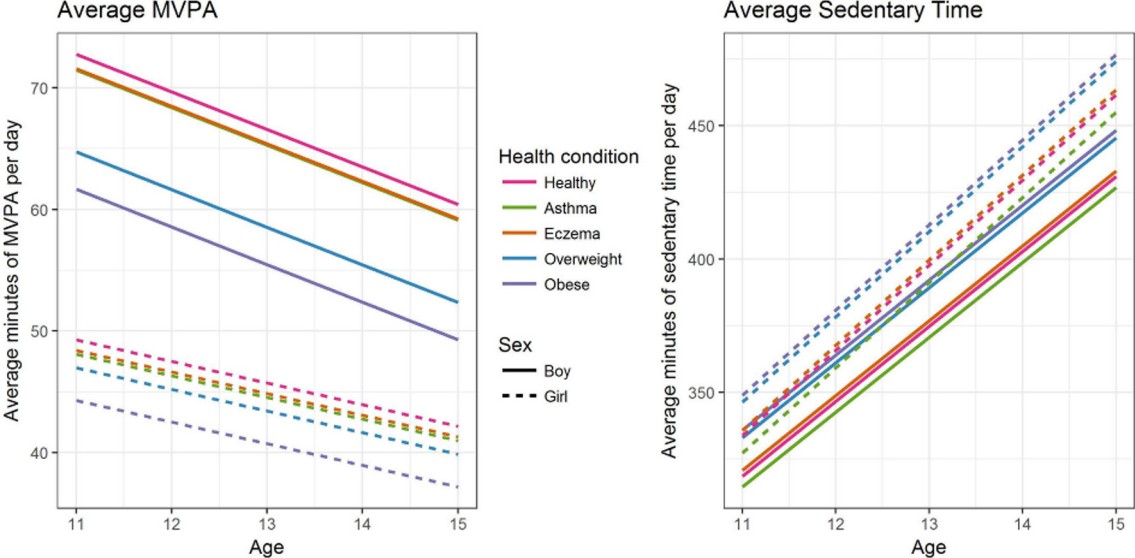

**Figure 1** Fitted models for average minutes of moderate to vigorous intensity physical activity (MVPA) (left) and sedentary time (right) for different long-term conditions and by sex.

among boys, and a decrease in MVPA of 5.0 min and an increase in sedentary time of 15.2 min among girls.

There was no evidence of any additional effect associated with having combinations of multiple long-term conditions, via interaction terms. This suggests that any associations between physical activity and long-term conditions may be additive, although low prevalence of multiple conditions in this sample mean that we lacked statistical power to explore this fully. There was no evidence that the rate of change in physical activity over time differed between health condition. Including puberty in the longitudinal model (results not shown) reduced the strength of association with obesity and being overweight slightly but had no effect on associations with asthma or eczema.

## DISCUSSION

In this paper, we found no strong evidence of differences in the mean minutes of MPVA or sedentary time per day between adolescents with asthma or eczema and those without these conditions in the cross-sectional models. We found a small association between asthma and fewer minutes of sedentary time among girls, but no associations for MVPA or eczema. Results suggest that adolescents who are overweight or obese engage in lower levels of physical activity at ages 12, 14 and 16, with stronger associations for boys than girls. These findings were confirmed in the longitudinal models which showed lower levels of MVPA and higher sedentary time with increasing levels of obesity. Specifically, obese boys engage in 11.1 fewer minutes of MVPA and 17.4 more sedentary minutes compared with a boy of similar age and demographics, while obese girls engage in 5.0 fewer minutes of MVPA and 15.2 more sedentary minutes than their non-obese counterparts. This finding is consistent with a previous Mendelian randomisation analysis which suggested a

likely bidirectional association between physical activity and adiposity among young people.[30]

The lack of an association between physical activity, sedentary time and asthma challenges a body of research which has reported that children with asthma are less active than their peers. However, previous literature has relied on relatively small cross-sectional studies using self-reported assessments of physical activity and asthma.[8 31–33] The current analysis uses a population-based rather than clinical cohort and therefore contains a lower proportion of children with moderate to severe asthma. Also, because the study combines data measured at different times, we report lifetime asthma prevalence rather than current asthma. Therefore, caution should be exercised in any direct comparison with other studies. Low physical activity and fitness have been suggested as risk factors for asthma onset[34] and exercise has been shown to improve markers of asthma control,[35] which appear inconsistent with our findings. Concerns that exercise-induced bronchoconstriction might limit physical activity in children with controlled asthma appear to be ill-founded.[36] Our results therefore provide reassurance that adolescents with asthma in a general population take part in similar levels of physical activity to their peers and there is no need for condition-specific behaviour change programmes.

A previous Mendelian randomisation study from the ALSPAC cohort at age 7 identified obesity as a risk factor for asthma.[37] This is consistent with our results. We observed a higher prevalence of obesity among asthma sufferers (20%, 16% and 17% at ages 12, 14 and 16) than those without asthma (14%, 12% and 12%, respectively). Furthermore, obesity with asthma has been implicated as a risk for more frequent asthma exacerbations,[38] condition severity and a reduced response to therapy.[39] Some small studies have suggested that weight loss may aid asthma management in children and adults.[40–42]

These studies and the consistent reporting of associations between asthma, obesity prevalence and condition morbidity supported by this study suggest that larger and longer weight management trials in obese individuals with asthma are warranted.

Previous evidence of any difference in the physical activity patterns of children with and without eczema was unclear.[9] There is some evidence of an association in adults,[43] and possibly an association with severe eczema in children.[44] There may also be a link between eczema and obesity in North America and Asia, although there is no evidence of an association in Europe.[45] Our analyses found no differences in accelerometer-measured physical activity for those with lifetime-diagnosed eczema of any severity, which is encouraging given the psychosocial impact of even mild eczema. While parents may restrict activities such as swimming because of concerns about the effects on the child's skin,[46] our findings suggest that the presence of eczema does not overall negatively impact physical activity in this group and therefore there is no need for eczema-specific physical activity programmes.

### Strengths and limitations

The main strength of this study is the objective assessment of physical activity at ages 12, 14 and 16 and multiple assessments of physical conditions on a large cohort of adolescents. Although only 25% participants had data at all three timepoints, 61% had data for at least two, making the longitudinal analysis reasonably robust, although the study was not powered to test for interactions between conditions. We used multiple imputation to minimise the amount of missing data but recognise that there are limitations of this approach, including the underlying assumption that data are missing at random.[47 48]

The study combines data measured at different times. Specifically, the asthma and eczema assessments do not exactly coincide with the physical activity data and so we were unable to include current asthma/eczema indicators. We also have no asthma/eczema information between 11 and 14 and so cannot determine, for example, if asthma precedes obesity. We have not adjusted for smoking we may have impacted on other associations. It is also important to highlight that due to the small number of cases, we were unable to model the interactions between health conditions. Finally, we are unable to differentiate between mild and severe asthma or eczema—more severe cases tend to be more persistent and it may be that persistence is linked to activity. The data were collected approximately a decade ago and it is possible that physical activity patterns and the prevalence of physical conditions, especially obesity, may have changed over this period.

### CONCLUSIONS

Analysis of the ALSPAC population-based cohort has shown that physical activity and sedentary time did not differ for adolescents with asthma or eczema and those without, but obese adolescents engaged in fewer minutes of MVPA and more sedentary time. Findings reinforce the need for strategies to help obese adolescents be more active but suggest no need to develop bespoke physical activity strategies for adolescents with mild asthma or eczema.

**Author affiliations**
[1]Centre for Exercise, Nutrition and Health Sciences, School for Policy Studies, University of Bristol, Bristol, UK
[2]National Institute for Health Research (NIHR) Collaboration for Leadership in Applied Health Research and Care West (CLAHRC West) at University Hospitals Bristol NHS Foundation Trust, Bristol, UK
[3]NIHR Biomedical Research Centre at the University Hospitals Bristol NHS Foundation Trust and the University of Bristol, Bristol, UK
[4]Bristol Dental School, University of Bristol, Bristol, UK
[5]Centre for Academic Primary Care, Population Health Sciences, Bristol Medical School, University of Bristol, Bristol, UK
[6]Population Health Sciences, Bristol Medical School, University of Bristol, Bristol, UK

**Acknowledgements** We are extremely grateful to all the families who took part in this study, the midwives for their help in recruiting them, and the whole ALSPAC team, which includes interviewers, computer and laboratory technicians, clerical workers, research scientists, volunteers, managers, receptionists and nurses.

**Contributors** Conception/design, revising and final approval: all authors. Methodology: RJ, RES, MJR, AJH, ARN, JPHS. Data analysis: RES. Drafting: RJ, RES.

**Funding** The UK Medical Research Council and Wellcome (Grant ref: 102215/2/13/2) and the University of Bristol provide core support for ALSPAC. This publication is the work of the authors and AJH, ARN, RES and RJ will serve as guarantors for the contents of this paper. A comprehensive list of grants funding is available on the ALSPAC website. Physical activity data collection was supported by the US National Heart, Lung, and Blood Institute (R01 HL071248-01A). RJ is partly supported by the National Institute for Health Research (NIHR) Collaboration for Leadership in Applied Health Research and Care West (CLAHRC West) at University Hospitals Bristol NHS Foundation Trust. The NIHR Biomedical Research Centre at University Hospitals Bristol NHS Foundation Trust and the University of Bristol supports ARN and JPH-S. MJR is funded by a National Institute for Health Research Post-Doctoral Research Fellowship (PDF-2014-07-013).

**Competing interests** None declared.

**Patient consent** Parental/guardian consent obtained.

**Ethics approval** ALSPAC Ethics and Law Committee and the Local Research Ethics Committees.

**Provenance and peer review** Not commissioned; externally peer reviewed.

**Data sharing statement** This is a secondary data analysis based on data from the ALSPAC cohort. The access policy for the ALSPAC data can be found at http://www.bristol.ac.uk/media-library/sites/alspac/documents/researchers/data-access/ALSPAC_Access_Policy.pdf.

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
