## [Reviewer comments · BMJ Open]

ARTICLE DETAILS

TITLE (PROVISIONAL)	ASSOCIATIONS BETWEEN PHYSICAL ACTIVITY AND ASTHMA, ECZEMA AND OBESITY IN CHILDREN AGED 12-16: AN OBSERVATIONAL COHORT STUDY
AUTHORS	Jago, Russ; Salway, Ruth; Ness, Andy; Shield, Julian; Ridd, Matthew; Henderson, A. John

VERSION 1 – REVIEW

REVIEWER	Kelly Morgan Cardiff University, Wales
REVIEW RETURNED	31-Jul-2018

GENERAL COMMENTS	Manuscript ID: bmjopen-2018-024858 The manuscript examines various long-term conditions and associations with moderate to vigorous physical activity and sedentary behaviour in 12-16 year olds. Data are taken from the ALSPAC study. The inclusion of objective physical activity and large sample size are strengths of the paper. Overall this paper is well written and with a few minor changes will provide a valuable addition to the growing body of research in this area. Please find my comments on the manuscript below. 1. Ensure consistency of terms. There is currently variability with 'long-term conditions', 'diseases' and 'health conditions' used throughout the paper.2. Add missing references to the sentence (p.5) 'There is some evidence that physical activity levels are lower among children with asthma than children without asthma'.3. The overall objective of the study (p.6) requires some attention to the phrasing 'and how they may change' please be more explicit as to what 'they' is referring to.4. Accelerometer data – please state whether accelerometers were waist or wrist worn and whether wear-time validation was considered5. Final paragraph of the results section (p.17) - first two sentences appear contradictory please revise or provide further details for clarity.6. Suggestion to amend 'This analysis ...' to 'The current analysis/study' for greater clarity (p.18, second paragraph) and insert 'cohort' after 'population-based'.
---

	7. Were the authors able to adjust for smoking within the models? As a significant confounder for asthma, if not possible, this at the very least should be included within the limitations section.
--	--

REVIEWER	Petr Baďura Faculty of Physical Culture, Palacký University Olomouc, Czech Republic
REVIEW RETURNED	14-Aug-2018

GENERAL COMMENTS	Review – bmjopen-2018-024858 Association between sitting, physical activity and asthma, eczema and obesity in children aged 12-16: An observational cohort study GENERAL COMMENT: The manuscript investigates both cross-sectional and longitudinal associations between physical activity (PA) and three most prevalent diseases in youth using a large rigorously collected dataset. Expectedly, the authors observed negative associations between PA with excessive weight. None of the associations with eczema and asthma was consistently significant across their sample. The authors present a large amount of information but I have to appreciate conciseness and clarity of their paper. Indeed, I read it with interest and especially liked the self-reflecting character of the Strengths and Limitations section, which facilitates proper understanding of their findings. However, I have a few questions and suggestions (mostly of minor nature) for revising the manuscript prior to its publication, as I miss some information in the Methods section. SPECIFIC COMMENTS:  1. Introduction (p. 5; l. 9-12) – The authors refer to age categories of 4-5 and 10-11-year-olds when reporting the prevalence rate of obesity in UK, however their sample is older, i.e. 12-16 years of age. In addition, they use terms ‘child/childhood’ in the Introduction and Methods section, however then they skip to ‘adolescents’ in the Aim of the study, Results and further. I would recommend clarifying that and ideally using the same terminology throughout the manuscript. 2. Introduction (p. 5; l. 46-48) – I am not a native speaker, but should not the text be ‘...if there was evidence of lower physical activity’ instead of ‘were’, i.e. singular as at the very beginning of the next paragraph. 3. Introduction (p. 6; l. 9) – Please define the ALSPAC abbreviation here, as it is the first mention in the text. 4. Methods (p. 7) – Although it is suggested in the Strengths and Limitations and it is possible to calculate it from mothers’ delivery dates, I believe it would be beneficial to state the years of data collection here. 5. Methods (p. 7; l. 24-31) – The authors state that they excluded adolescents with less than three days of valid data. However, I wonder if they considered differences in PA between weekday and weekends, as sometimes combined exclusion criteria are used – at least X weekdays and X weekend days. If seven days of accelerometer monitoring were consecutive, please indicate so. Did you take eventual effect of reactivity into account? 6. Methods (p. 7; l. 40-44) – The authors mention 21 cases were excluded from the analyses. I believe it means 21 respondents, but it could also be understood as 21 days of monitoring. Please clarify.
---

	7. Methods (p. 7; l. 50-55) – Were height and weight measured in a clinical setting by a physician or in other way? Please specify. 8. Methods (p. 7; l. 55-57) – ‘Throughout this paper the overweight category comprises adolescents who were overweight but not obese’. I see the point, however find the sentence somewhat confusing and misleading in the sense that obese adolescents were not of interest. Please reword the sentence. 9. Methods (p. 8; l. 22-23) – ‘Mother’s highest education was combined from...’ I am not sure that I understand what was meant by the word ‘combined’, could you please clarify. 10. Methods (p. 8; l. 48-51) – ‘Asthma and eczema diagnoses....., so were taken from the closest prior measurement’. Was not it supposed to be lifetime manifestation of these diseases? 11. Results (p. 11; l. 20-22) – The authors write that Table S3 summarises MVPA/sedentary time of adolescents with any of three diagnoses and those without health condition. However, the table actually only presents data on ‘diagnosed’ adolescents and comparison with the ‘healthy’ ones. 12. Results (p. 11; l. 43-44) – ‘MVPA was 3.1-7.5 minutes lower...’ Table 1 shows difference (non-significant though) of -2.27 in 16-year-olds. 13. Results (p. 11; l. 48-51) – ‘These associations tend to be slightly stronger in boys than girls’. This does not seem to be exact wording, as maybe even more importantly, 5 out of 6 differences were significant in boys, while ‘only’ 3 in girls, where no association with weight status was found among girls aged 16. 14. Table 2 (p.14) – In footnote 1, there is a different size of font used. 15. Results (p. 11; l. 42) – The authors mention that inclusion puberty as a confounder, did not markedly affect the associations. Would it be possible to add the data as a supplementary material, similarly to ‘minimally adjusted’ models? 16. Where appropriate, could you please add units (min) to tables in the body of the text, as well as in supplementary materials?
--	---

REVIEWER	Ma Jun Institute of Child and Adolescent Health of Peking University, China
REVIEW RETURNED	16-Aug-2018

GENERAL COMMENTS	This paper aimed to explore the association between physical activity and three most common long-term conditions in UK children. The study was a part of the Avon Longitudinal Study and of good design, especially for the accurate measurement of physical activity, which is quite commendable. This study clearly demonstrated the association between physical activity time and overweight/obese, eczema and asthma conditions separately, however, as these three conditions were treated as a whole as “the most common long-term conditions for UK children”, the combined effects also need report in the main text, or it would be little point to combine these three condition in one single paper. Previous studies had found that these three conditions could have influence on each other and may have joint effect on children health, whether the interaction on physical activity exist or not could be meaningful in practice work, thus there may need more space for the interaction part of results and discussion.
--

VERSION 1 – AUTHOR RESPONSE

Thank you for the helpful review of our paper. Please find below a point by point response to all issues raised. Changed text is shown in yellow in the revised paper.

Reviewer: 1

Reviewer Name: Kelly Morgan

Institution and Country: Cardiff University, Wales

Please state any competing interests or state 'None declared': None declared

The manuscript examines various long-term conditions and associations with moderate to vigorous physical activity and sedentary behaviour in 12-16 year olds. Data are taken from the ALSPAC study. The inclusion of objective physical activity and large sample size are strengths of the paper. Overall this paper is well written and with a few minor changes will provide a valuable addition to the growing body of research in this area. Please find my comments on the manuscript attached.

1. Ensure consistency of terms. There is currently variability with 'long-term conditions', 'diseases' and 'health conditions' used throughout the paper.

Response: *The text has been changed to long-term conditions throughout.*

2. Add missing references to the sentence (p.5) 'There is some evidence that physical activity levels are lower among children with asthma than children without asthma'.

Response: *Reference has been added. Thanks for catching.*

3. The overall objective of the study (p.6) requires some attention to the phrasing 'and how they may change' please be more explicit as to what 'they' is referring to.

Response: *We have now re-phrased this objective to "The overall objective of this paper is to examine the physical activity of adolescents with three of the most common long-term conditions and if these associations change as young people move through adolescence, using objective assessments of physical activity". Please see page 6.*

4. Accelerometer data – please state whether accelerometers were waist or wrist worn and whether wear-time validation was considered

Response: *We have now clarified that the monitor was worn on the hip for seven days and that a valid day was 500 minutes of data. Please see page 7.*

5. Final paragraph of the results section (p.17) - first two sentences appear contradictory please revise or provide further details for clarity.

Response: *We have changed the text to: 'There was no evidence of any additional effect associated with having combinations of multiple long-term conditions, via interaction terms.'*

6. Suggestion to amend 'This analysis ...' to 'The current analysis/study' for greater clarity (p.18, second paragraph) and insert 'cohort' after 'population-based'.

Response: *Change made. Please see page 17.*

7. Were the authors able to adjust for smoking within the models? As a significant confounder for asthma, if not possible, this at the very least should be included within the limitations section.

Response: *We have now added this as a limitation. Please see page 19.*

Reviewer: 2

Reviewer Name: Petr Baďura

Institution and Country: Faculty of Physical Culture, Palacký University Olomouc, Czech Republic

Please state any competing interests or state 'None declared': None declared

GENERAL COMMENT:

The manuscript investigates both cross-sectional and longitudinal associations between physical activity (PA) and three most prevalent diseases in youth using a large rigorously collected dataset. Expectedly, the authors observed negative associations between PA with excessive weight. None of the associations with eczema and asthma was consistently significant across their sample. The authors present a large amount of information but I have to appreciate conciseness and clarity of their paper. Indeed, I read it with interest and especially liked the self-reflecting character of the Strengths and Limitations section, which facilitates proper understanding of their findings. However, I have a few questions and suggestions (mostly of minor nature) for revising the manuscript prior to its publication, as I miss some information in the Methods section.

Response: *Thanks for the supportive comments. Please see responses below.*

SPECIFIC COMMENTS:

1. Introduction (p. 5; l. 9-12) – The authors refer to age categories of 4-5 and 10-11-year-olds when reporting the prevalence rate of obesity in UK, however their sample is older, i.e. 12-16 years of age. In addition, they use terms 'child/childhood' in the Introduction and Methods section, however then they skip to 'adolescents' in the Aim of the study, Results and further. I would recommend clarifying that and ideally using the same terminology throughout the manuscript.

Response: *We have used the term child and childhood to describe the studies in the background evidence as that is the best available evidence. The current study focusses on adolescence and as such we believe the text is correct and does not need to be changed.*

2. Introduction (p. 5; l. 46-48) – I am not a native speaker, but should not the text be '...if there was evidence of lower physical activity' instead of 'were', i.e. singular as at the very beginning of the next paragraph.

Response: *We believe the text is correct and so have not made a change.*

3. Introduction (p. 6; l. 9) – Please define the ALSPAC abbreviation here, as it is the first mention in the text.

Response: *Change has been made. Please see page 6.*

4. Methods (p. 7) – Although it is suggested in the Strengths and Limitations and it is possible to calculate it from mothers' delivery dates, I believe it would be beneficial to state the years of data collection here.

Response: *We have added the following to the accelerometer section on page 7: "The first accelerometer measurement was taken at an average age of 11 years and 9 months in 2003-4, with subsequent measurements at ages 13 years and 10 months, and at 15 years and 6 months."*

5. Methods (p. 7; l. 24-31) – The authors state that they excluded adolescents with less than three days of valid data. However, I wonder if they considered differences in PA between weekday and weekends, as sometimes combined exclusion criteria are used – at least X weekdays and X weekend days. If seven days of accelerometer monitoring were consecutive, please indicate so.

Response: *The analysis is based on 3 valid days, rather than 2 weekday /1 weekend. As the focus of the analyses was on habitual physical activity we have not conducted additional analyses by weekday and weekend day.*

6. Methods (p. 7; l. 40-44) – The authors mention 21 cases were excluded from the analyses. I believe it means 21 respondents, but it could also be understood as 21 days of monitoring. Please clarify.

Response: *You are correct – we have changed this to respondents to clarify.*

7. Methods (p. 7; l. 50-55) – Were height and weight measured in a clinical setting by a physician or in other way? Please specify.

Response: *Height as weight were measured in a clinic by trained Fieldworkers. This has been added to page 7.*

8. Methods (p. 7; l. 55-57) – ‘Throughout this paper the overweight category comprises adolescents who were overweight but not obese’. I see the point, however find the sentence somewhat confusing and misleading in the sense that obese adolescents were not of interest. Please reword the sentence.

Response: *Text has been changed and now reads as “Throughout this paper the overweight category is the participants who were overweight but not the participants who were obese, who are defined as obese throughout.”*

9. Methods (p. 8; l. 22-23) – ‘Mother’s highest education was combined from...’ I am not sure that I understand what was meant by the word ‘combined’, could you please clarify.

Response: *Text has been changed to read “The mother’s highest education variable was created by combining from data at 32w gestation and age 8 to provide the highest education recorded in the dataset.”*

10. Methods (p. 8; l. 48-51) – ‘Asthma and eczema diagnoses....., so were taken from the closest prior measurement’. Was not it supposed to be lifetime manifestation of these diseases?

Response: *This is lifetime diagnosis up to the accelerometer measurement – so someone diagnosed between age 12 and 14 will not have asthma at age 12 for that part of the analysis, but will have asthma in the subsequent age 14 & 16 analyses. We have added ‘... to record lifetime diagnosis up to the point of the accelerometer measurement.’*

11. Results (p. 11; l. 20-22) – The authors write that Table S3 summarises MVPA/sedentary time of adolescents with any of three diagnoses and those without health condition. However, the table actually only presents data on ‘diagnosed’ adolescents and comparison with the ‘healthy’ ones.

Response: *We have clarified this to read ‘the difference in MVPA and sedentary time for adolescents with asthma, eczema, overweight and obese conditions, compared to those without.’*

12. Results (p. 11; l. 43-44) – ‘MVPA was 3.1-7.5 minutes lower...’ Table 1 shows difference (non-significant though) of -2.27 in 16-year-olds.

Response: *We have corrected this to read ‘2.3-7.5 minutes lower’ to reflect the numbers in the table. However, in line with ALSPAC analysis guidelines we do not draw attention to significance/non-significance as this is an observational study.*

13. Results (p. 11; l. 48-51) – ‘These associations tend to be slightly stronger in boys than girls’. This does not seem to be exact wording, as maybe even more importantly, 5 out of 6 differences were significant in boys, while ‘only’ 3 in girls, where no association with weight status was found among girls aged 16.

Response: *The original text was specifically worded to focus on the nature of the associations as opposed to a focus on statistical significance. We have adopted this policy as we want to ensure that all text does not overly focus on results because a p-value is one side or another of the 0.05 threshold. This is also the ALSPAC publication policy. We therefore believe that the text is appropriate and we have not made any changes.*

14. Table 2 (p.14) – In footnote 1, there is a different size of font used.

Response: *Change made. Thanks for catching.*

15. Where appropriate, could you please add units (min) to tables in the body of the text, as well as in supplementary materials?

Response: *Units have been added to the headers*

Reviewer: 3

Reviewer Name: Ma Jun

Institution and Country: Institute of Child and Adolescent Health of Peking University, China

Please state any competing interests or state 'None declared': none declared

This paper aimed to explore the association between physical activity and three most common long-term conditions in UK children. The study was a part of the Avon Longitudinal Study and of good design, especially for the accurate measurement of physical activity, which is quite commendable. This study clearly demonstrated the association between physical activity time and overweight/obese, eczema and asthma conditions separately, however, as these three conditions were treated as a whole as "the most common long-term conditions for UK children", the combined effects also need report in the main text, or it would be little point to combine these three condition in one single paper. Previous studies had found that these three conditions could have influence on each other and may have joint effect on children health, whether the interaction on physical activity exist or not could be meaningful in practice work, thus there may need more space for the interaction part of results and discussion.

Response: *We have outlined in the text at the top of page 15 that we explored interactions between long-term conditions but due to the small number of cases it was not possible to model these in the analyses. We have also flagged this as a limitation on page 20.*

FORMATTING AMENDMENTS (if any)

Required amendments will be listed here; please include these changes in your revised version:

- Kindly re-upload each figure under 'Image' file designation with at least 300 dpi resolution and at least 90mm x 90mm of width.

Response: *High resolution figures have been provided.*

- Please re-upload your supplementary files in PDF format.

Response: *PDF version of supplementary file have been provided.*

- We have implemented an additional requirement to all articles to include 'Patient and Public Involvement' statement within the main text of your main document. Please refer below for more information regarding this new instruction:

Authors must include a statement in the methods section of the manuscript under the sub-heading

'Patient and Public Involvement'.

This should provide a brief response to the following questions:

How was the development of the research question and outcome measures informed by patients' priorities, experience, and preferences?

How did you involve patients in the design of this study?

Were patients involved in the recruitment to and conduct of the study?

How will the results be disseminated to study participants?

For randomised controlled trials, was the burden of the intervention assessed by patients themselves?

Patient advisers should also be thanked in the contributorship statement/acknowledgements.

If patients and or public were not involved please state this.

Response: *We have added new PPI text to page 8.*

VERSION 2 – REVIEW

REVIEWER	Petr Baďura Palacký University Olomouc, Czech Republic
REVIEW RETURNED	25-Oct-2018
GENERAL COMMENTS	I would like to thank the authors for revising their manuscript according to suggestions of mine and the other two reviewers. All my comments were either implemented to the paper or explained to my full satisfaction, except for the first one. It is exactly as written in the 'JH2' comment. I would just expect to find also the information about prevalence of obesity in the age category corresponding to the age of your sample (12–16 years). However, this is a very minor remark and I recommend the manuscript to be published in BMJ Open as I believe it can contribute to the growing body of studies in the field.